# Dichotomous organization of amygdala/temporal-prefrontal bundles in both humans and monkeys

**Davide Folloni[1,2]\*, Jerome Sallet[1,2], Alexandre A Khrapitchev[3], Nicola Sibson[3], Lennart Verhagen[1,2,4†], Rogier B Mars[2,4†]**

[1]Wellcome Centre for Integrative Neuroimaging (WIN),Department of Experimental Psychology, University of Oxford, Oxford, United Kingdom; [2]Wellcome Centre for Integrative Neuroimaging (WIN), Centre for Functional MRI of the Brain (FMRIB), Nuffield Department of Clinical Neurosciences, John Radcliffe Hospital, University of Oxford, Oxford, United Kingdom; [3]Department of Oncology, Cancer Research UK and Medical Research Council Oxford Institute for Radiation Oncology, University of Oxford, Oxford, United Kingdom; [4]Donders Institute for Brain, Cognition and Behaviour, Radboud University Nijmegen, Nijmegen, Netherlands

**Abstract** The interactions of anterior temporal structures, and especially the amygdala, with the prefrontal cortex are pivotal to learning, decision-making, and socio-emotional regulation. A clear anatomical description of the organization and dissociation of fiber bundles linking anterior temporal cortex/amygdala and prefrontal cortex in humans is still lacking. Using diffusion imaging techniques, we reconstructed fiber bundles between these anatomical regions in human and macaque brains. First, by studying macaques, we assessed which aspects of connectivity known from tracer studies could be identified with diffusion imaging. Second, by comparing diffusion imaging results in humans and macaques, we estimated the patterns of fibers coursing between human amygdala and prefrontal cortex and compared them with those in the monkey. In posterior prefrontal cortex, we observed a prominent and well-preserved bifurcation of bundles into primarily two fiber systems—an amygdalofugal path and an uncinate path—in both species. This dissociation fades away in more rostral prefrontal regions.
DOI: https://doi.org/10.7554/eLife.47175.001

**\*For correspondence:** davide.folloni@psy.ox.ac.uk

[†]These authors contributed equally to this work

**Competing interests:** The authors declare that no competing interests exist.

## Introduction

The neural circuits centered on the amygdala and temporal lobe on the one hand and the prefrontal cortex on the other are crucial in a variety of complex behaviors, including reward-based learning and decision-making (*De Martino et al., 2006*; *Hunt and Hayden, 2017*; *Morrison and Salzman, 2010*; *Murray and Wise, 2010*; *Rudebeck and Murray, 2014*; *Rushworth et al., 2011*), and emotional and social behavior (*Klüver and Bucy, 1937*; *Weiskrantz, 1956*; *Noonan et al., 2014*; *Volman et al., 2011*; *Whalen et al., 1998*). Recent models increasingly emphasize how the amygdala, a complex subcortical area located in the medial bank of the anterior temporal lobe, and the prefrontal cortex (PFC) do not support isolated computations but instead have complementary roles in these processes (*Munuera et al., 2018*; *Saez et al., 2015*; *Murray and Wise, 2010*). Similar computational divisions of labor have been proposed for the anterior temporal association cortex and the prefrontal cortex (*Freedman et al., 2003*).

Understanding these computational roles requires an understanding of the anatomy of the brain systems that are involved (*Marr, 1982*). Specifically, we want to understand the principles of connectivity between amygdala/anterior temporal systems and prefrontal systems, in order to better

comprehend how information can flow between the various nodes of these networks. Such principles, for instance, are evident in early models by *Carmichael and Price (1995)* that argue for a mostly dichotomous organization of circuits stretching across temporal and frontal lobes. This line of work, however, has mostly been based on invasive tracer studies that measure area-to-area connections with high spatial accuracy. These approaches are not available for in-vivo human studies. Recent work on diffusion MRI tractography, the only available method to establish structural connectivity in-vivo, has demonstrated that using this approach to reconstruct area-to-area connections has many pitfalls (*Maier-Hein et al., 2017*; *Reveley et al., 2017*). A more feasible strategy is to reconstruct fiber bundles (*Mars et al., 2016a*; *Thiebaut de Schotten et al., 2011*). By seeding in the reliably identifiable white matter, it becomes possible to isolate the core of fiber bundles and then follow their trajectory towards the gray matter in both directions. Although this technique has the potential to elucidate principles of connectivity between systems, it relies on a different philosophy than the invasive tracer studies, making it difficult to draw firm comparative conclusion between results obtained using the two techniques. Therefore, we here apply this technique first in the macaque monkey, for which tracer results are available and thus comparison of the results of the two techniques is possible, and then to the human.

White matter dissections and tract-tracing studies in non-human primates have shown that anterior temporal regions, including amygdala, and the frontal lobe have widespread reciprocal connections (*Aggleton et al., 2015*; *Barbas et al., 2011*; *Barbas and De Olmos, 1990*; *Ghashghaei et al., 2007*; *Ghashghaei and Barbas, 2002*; *Price, 2003*; *Timbie and Barbas, 2014*). Two primary pathways are known to connect anterior temporal regions including the amygdala and the frontal lobe: a major association fiber pathway, the uncinate fascicle (UF) (*Amaral and Price, 1984*; *Déjerine, 1895*; *Klingler and Gloor, 1960*; *Kuypers et al., 1965*; *Nauta, 1961*; *Thiebaut de Schotten et al., 2012*; *Yeterian et al., 2012*), and a limbic-cortical ventral amygdalofugal pathway (from here on also referred simply to as the amygdalofugal pathway, AmF) (*Aggleton et al., 1980*; *Bachevalier et al., 1985*; *Lehman et al., 2011*; *Nauta, 1961*; *Nieuwenhuys et al., 2008*; *Russchen et al., 1985*). To date, the AmF has not been described in great detail in the primate brain in-vivo (*Carmichael and Price, 1995*; *Croxson et al., 2005*) and is often omitted from human studies of circuits for social-emotional behavior and decision making (*Alm et al., 2015*; *Ameis and Catani, 2015*), even though its role in cognitive and emotional processes in animals is critical (*Bachevalier et al., 1985*; *de Guglielmo et al., 2019*). Therefore, we focus our investigation on the translatability of knowledge of macaque amygdala/anterior temporal-prefrontal fiber bundles to the human brain along these two tracts.

We used high-resolution diffusion MRI data in the macaque to ascertain the degree to which the diffusion MRI tractography approach identified aspects of amygdala/anterior temporal-prefrontal connectivity known from previous macaque studies. This demonstrated that the core bundles of the AmF perforate the porous substantia innominata in the basal forebrain. In this inhomogeneous tissue, the fractional anisotropy index is not fully informative of tract integrity. This could lead to false negatives in deterministic tractography, but the probabilistic tractography approach that we employ here demonstrates a distribution of streamlines running through this area. Accordingly, here we have adopted a probabilistic approach to tractography that is strongly informed by prior anatomical knowledge to reconstruct this tract in a manner similar to that used in previous tract tracing studies.

Using this approach, we were able to robustly define and to dissociate a medial amygdalofugal pathway and an orbital uncinate tract in both the macaque and human brain. Together, these pathways form a larger constellation of amygdala/anterior temporal-prefrontal circuits, but each with a distinct connectional profile interfacing with a unique set of brain regions. While the amygdalofugal pathway predominantly ran in the white matter exntending between the amygdala, nucleus accumbens, and subgenual cingulate, ventromedial, and frontopolar regions in ventral PFC, the uncinate pathway primarily coursed in the white matter adjacent to anterior temporal regions, including the amygdala, lateral orbital areas and frontopolar cortex. The relationship and structure of these tracts is preserved across primate species, supporting the translation of insights from non-human primate anatomy to inform our understanding of the human brain.

## Results

Our goal was to reconstruct the anatomical organization of the amgydalofugal and uncinate bundles in white matter (WM) within amygdala/anterior temporal regions (ATR) and prefrontal territory. In particular, we were interested in assessing the course of these pathways towards the prefrontal cortex and their interaction with a wide range of prefrontal regions of interest. Given the difficulty of reconstructing these pathways, we employed the following strategy. First, we reconstructed the tracts in the macaque monkey brain, so that they can be compared to known tracer results (*Figure 1A–C*, left and middle panels). This demonstrated whether our tractography techniques are valid for these pathways. Then, we applied the same tracking strategy to the human, using the seeding and masking procedures applied successfully in the macaque (*Figure 1A–C*, right panels). Finally, we compared the tracts' projections directly, across species and with respect to a number of control tracts.

### Amygdalofugal pathway in the macaque monkey brain

The frontal body of the AmF was represented by a thin bundle of fibers running in the WM adjacent to medial subcortical and cortical brain structures (*Figure 2A–E* and *Figure 2—figure supplement 1A–C*). In a posterior section of the macaque brain, fibers running in the medially projecting limb of the AmF pathway coursed in the WM dorsal to the amygdala, lying between the amygdala and the medial longitudinal fissure at the level of the posterior anterior commissure (AC) and the sublenticular extended amygdala area (SLEA) (*Figure 1A–B*, *Figure 2A* and *Figure 2—figure supplement 1A–B*). A first bundle of AmF fibers ran in the WM territory in between the nucleus basalis of Meynert, the lentiform nucleus, the ventral pallidum, and the piriform cortex (*Figure 2—figure supplement 1A–B*). The medial AmF fibers are known to separate into different sets of projections (*Lehman et al., 2011*; *Nauta, 1961*; *Novotny, 1977*; *Oler et al., 2017*) that extended ventrally to innervate the anterior hypothalamic areas at the level of the paraventricular-supraoptic nuclei and dorsally to connect the amygdala with the thalamus, the BNST, and the septum. All of these fibers in our data visibly left the seed location, but the frontal waypoint mask means that the pathways to the prefrontal cortex, which was the focus of the current investigation, were the most prominent (*Figure 2A* and *Figure 2—figure supplement 1A–B*). A second bundle of the AmF ran in the WM surrounding the substantia innominata, ventral pallidum, and ventral bank of the AC, and more rostrally in the region ventral to the ventral striatum/nucleus accumbens (VS/NAc; *Figure 1C* and *Figure 2A–E*). From tract tracing, we know that some fibers connecting ventral PFC and amygdala are intermingled with the much more prominent anterior commissure (*Lehman et al., 2011*). The AC is very prominent in the principal fiber direction of diffusion MRI data, but nevertheless some weak AmF fibers overlapping with AC were evident in our tractography data (*Figure 1B*, *Figure 2* and *Figure 2—figure supplement 1B*).

It must be noted that over its course in the basal forebrain and ventral to the striatum, the AmF is likely to interdigitate with other connectional systems, including the rostral fibers of the thalamic projections, the medial forebrain bundle, the diagonal band of Broca and the dopaminergic fibers connecting the ventral tegmental area with prefrontal cortex. Eventually, the anterior fibers of the AmF extended into the orbital periallocortex on the medial wall of the hemisphere and into the PFC by coursing in WM adjacent to the subgenual anterior cingulate cortex (sACC or Brodmann area 25; *Figure 2A,B,D* and *Figure 2—figure supplement 1C*).

Near the sACC, the AmF split again into two sub-sections. The first sub-section ran dorsally alongside the cingulum bundle (CB) in anterior cingulate sulcus and perigenual WM. The second branch instead stretched between the sACC and the frontopolar cortex, passing subjacent to medial orbitofrontal cortex (OFC) between the gyrus rectus and the medial orbital gyrus (*Figure 2B* and *Figure 2D*). Because diffusion MRI cannot discern between monosynaptic and polysynaptic connections, the identified fibers could represent either type of projections and could run in either direction.

### Amygdalofugal pathway in the human brain

As in the macaque, we observed that in the human brain the AmF pathway ran adjacent to the lateral and basal nuclei of the amygdala, and curves in the more dorsal WM within the substantia innominate and SLEA bordering the central and medial nuclei, where additional fibers entered the

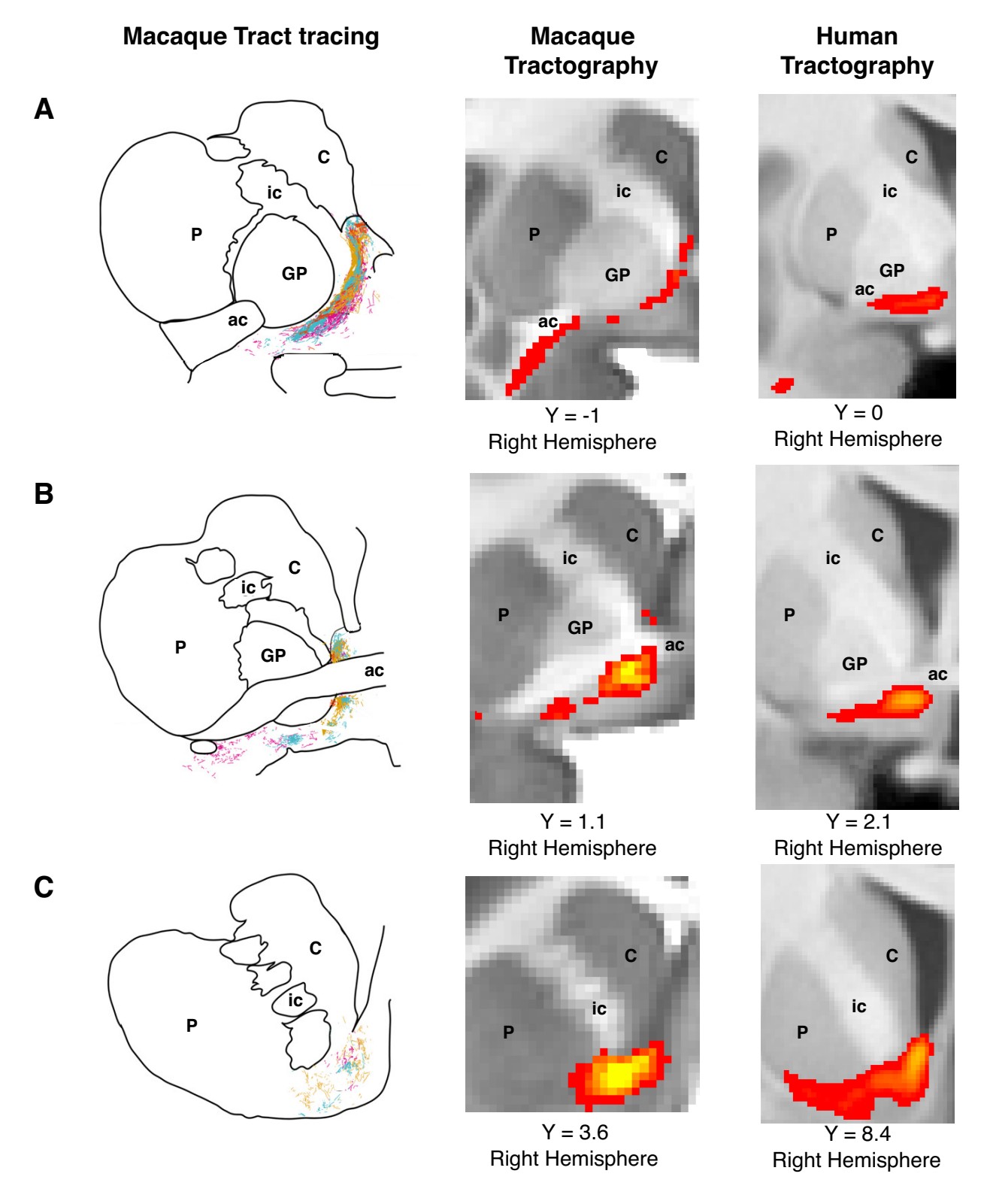

**Figure 1.** Histology-informed approach for the reconstruction of the frontal limb of the ventral amygdalofugal pathway. (**A-C**, left and middle panels) Coronal sections depicting reconstructions of the frontal limb of the amygdalofugal pathway (AmF) in the macaque brain using probabilistic tractography based on diffusion MRI data (middle panels) and published reconstructions of the tract using radio-labeled fibers identified using tract-tracing injections in macaque (left panels; adapted with permission from Springer Nature (*Oler et al., 2017*); These panels are not available under CC-

*Figure 1 continued on next page*

*Figure 1 continued*

BY and are exempt from the CC-BY 4.0 license.) The AmF tractography reconstruction showed marked correspondence in anatomy and in the course of the fibers with the histological tracing data. AmF fibers leave the amygdala caudal to the anterior commissure (ac) and turn medially towards the medial wall of the hemisphere (A) and rostrally towards the frontal lobe by coursing ventrally to the ventromedial striatum (B–C). (A-C, right panels) The histology-informed reconstruction of the AmF was then used to guide the tractographic algorithm in the human brain, including when the seed, exclusion and waypoint masks were used in both species (see Materials and methods for details on the masks). The human AmF follows a course matching the results observed in the macaque brain (A-C, middle panels) with fibers projecting both medially (A) and rostrally (B–C). T1-weighted images are shown in radiological convention. ac, anterior commissure; C, caudate nucleus; GP, globus pallidus; ic, internal capsule; P, putamen. © 2017 Springer Nature. All Rights Reserved. *Figure 1A–C* (left panels) is adapted with permission from Springer Nature (*Oler et al., 2017*).
DOI: https://doi.org/10.7554/eLife.47175.002

main body of the tract (*Figure 1A–C* right panels, *Figure 2A* and *Figure 2—figure supplement 1A–B*). Here, the AmF extended subcortically along the medial wall and gave rise to two limbic bundles. As in the macaque, the first set of AmF fibers constituted a medially projecting path that followed a trajectory resembling a reversed S-shape coursing in the WM between the caudal amygdala, periventricular regions both superior and inferior to the AC, and bed nuclei of stria terminalis, hypothalamus and thalamus. A second portion of the ventral AmF pathway coursed in the WM surrounding the substantia innominata and SLEA by interdigitating with AC fibers or by running through the substriatal WM surrounding the VS/NAc (*Figure 2A* and *Figure 2—figure supplement 1A–B*).

The human AmF in the PFC predominantly coursed alongside the sACC, although a smaller contingent of fibers bordered the caudal OFC WM. In ventral PFC, the main body of the AmF continued to run along the medial bank of the cortex in a position adjacent to ventromedial PFC (vmPFC) in the gyrus rectus and medial OFC (*Figure 2A*, *Figure 2C*, *Figure 2E* and *Figure 2—figure supplement 1C*). Recently, *Neubert et al. (2014)* suggested that the frontopolar cortex (FPC) in the human brain is composed of functionally specialized medial and lateral sub-regions to a greater extent than in the macaque. The AmF projections continued to run close to the medial wall in the FPC but a smaller number of projections also extended laterally to reach the lateral FPC (*Figure 2C* and *Figure 2E*).

The architecture of the human AmF closely resembled that of the macaque brain, suggesting that this tract is evolutionarily preserved; both its medial and anterior branches were very similar in the human and macaque monkey brains (*Figure 2A–E*) despite the limited WM in the latter species (*Schoenemann et al., 2005*). Minor differences arose when we look at FPC; in the human brain, few AmF fibers headed towards the lateral FPC and the majority of the tract innervated medial FPC (*Figure 2C and E*).

## Uncinate fascicle in the macaque monkey brain

The UF is a hook-shaped tract containing fibers that connect the temporopolar cortex and amygdala with orbital gyri and ventrolateral PFC in a bidirectional manner. On a ventral section, the macaque UF extended along the lateral surface of the temporal lobe where it made a dorsal C-shaped curve through the superior temporal gyrus of the temporal lobe (*Figure 2D* and *Figure 2—figure supplement 1D*). Although the principal fiber direction in this region runs in a superior-inferior plane, some fibers branched along a medial-lateral axis towards the amygdala, thereby interconnecting fibers adjacent to ATR and lateral prefrontal cortex (*Figure 2A,D* and *Figure 2—figure supplement 1E*). Other fibers also ran in a more caudal position along the parahippocampal gyrus and rhinal cortex (*Figure 2D* and *Figure 2—figure supplement 1D–E*). At the level of the limen insulae, UF fibers ran in a WM region ventrally to the claustrum, ventrolaterally to the putamen and globus pallidus, and medially to the insular cortex. Here, the most dorsal fibers of the UF intermingled with the ventral axons of the EmC as well as with ventrolateral PFC WM (*Figure 2A* and *Figure 2—figure supplement 1D–F*).

In monkey PFC, the UF strongly innervated the WM bordering the caudolateral OFC and the opercular and inferior frontal areas, although a sub-section of fibers extended next to the subgenual cingulate area and the medial bank of posterior OFC (*Figure 2A–B* and *Figure 2—figure supplement 1F*). Here, the more medial UF fibers joined with the AmF and CB fibers to innervate the ACC WM (*Figure 2B*). Overall, the main body of the UF in ventral PFC stretched throughout lateral,

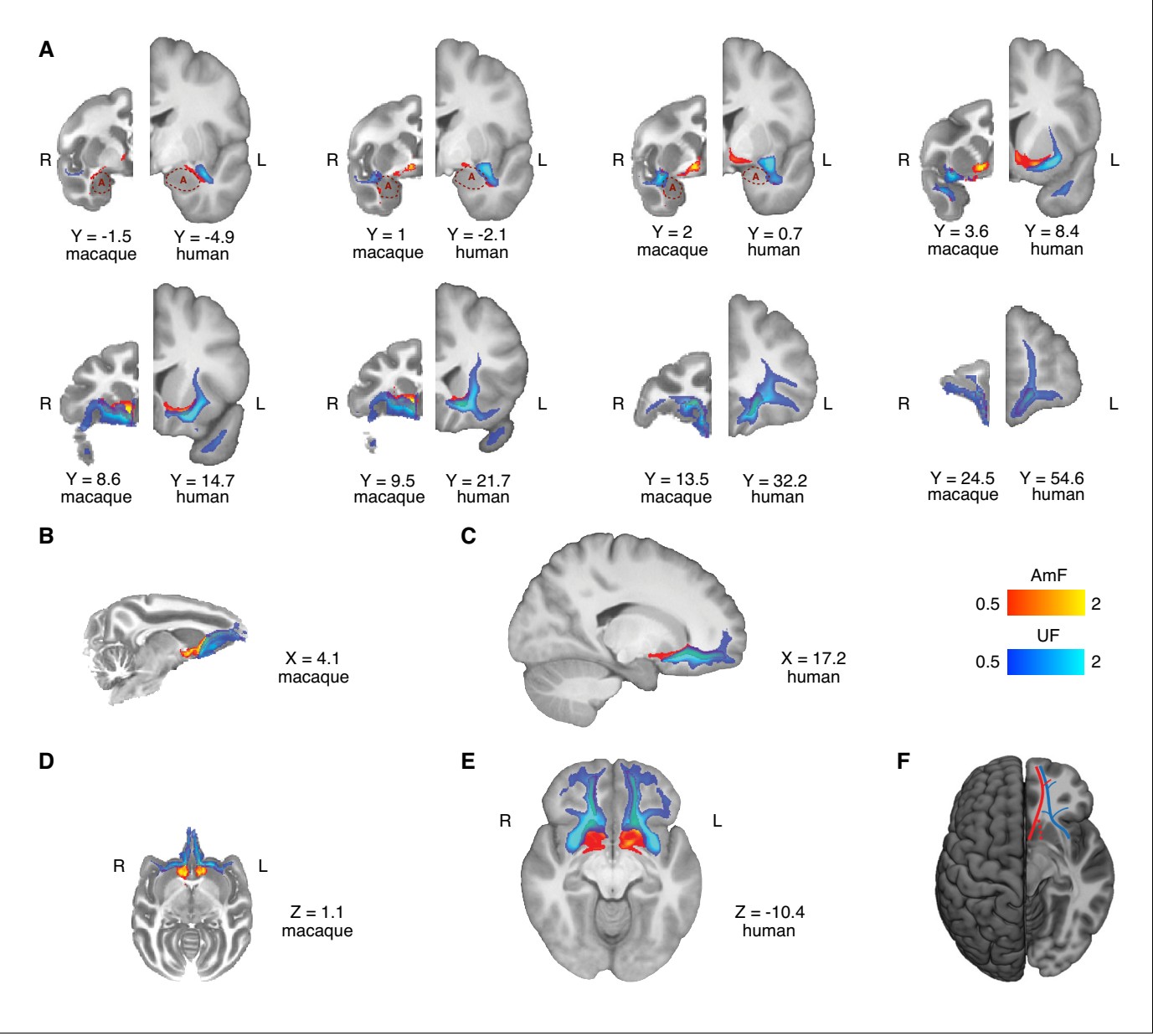

**Figure 2.** Anatomy and organization of the amygdalofugal pathway and uncinate fascicle in the basal ganglia and frontal lobe of the macaque monkey and human brain. (A) The amygdalofugal pathway (AmF, red) and uncinate fascicle (UF, blue) are displayed on eight coronal brain slices. AmF and UF show the same medial-lateral organization in both macaque (left) and human (right) brain. (B–C) Sagittal view of the macaque (B) and human (C) brains displaying the preserved dorsal-ventral and medial-lateral organization of the two tracts across both species in caudal subcortical regions of the brain and their merging in caudal orbitofrontal cortex white matter (WM). (D–E) The medial-lateral organization of the two tracts is visible on axial sections of macaque (D) and human (E) brains. The AmF follows a course medial to the UF with its fibers reaching the full extent of the ventral striatum and nucleus accumbens in both macaques (D) and humans (E). Here, the main body of the UF runs more laterally, primarily in the WM adjacent to the ventral-lateral striatum and the insular cortex. The tracts merge into an intermingled connectional system in the posterior OFC (area 13) WM and across the full extent of the ventromedial prefrontal cortex. Macaque coordinates are in F99 space and human coordinates are in MNI space. (F) Schematic template representing the organization of AmF (red) and UF (blue) in the ventral prefrontal cortex in the human brain. The dashed line (red) shows a bundle subcortical AmF fibers running underneath and through the ventral striatum.

DOI: https://doi.org/10.7554/eLife.47175.003

The following figure supplement is available for figure 2:

**Figure supplement 1.** Comparison of the anatomical architecture of the ventral amygdalofugal pathway and uncinate fascicle in macaque monkey and human brains.
DOI: https://doi.org/10.7554/eLife.47175.004

central, and medial orbital gyri, giving rise to a complex network in OFC which extended into the frontopolar cortex (*Figure 2D*).

## Uncinate fascicle in the human brain

The human UF had a preserved C-shaped structure lying between rostromedial temporal regions and ventrolateral PFC (*Figure 2C* and *Figure 2—figure supplement 1D*). Similarly to macaque temporal UF, it was anatomically organized predominantly along a dorsal-ventral gradient but with lateral fibers extending towards amygdalar nuclei and parahippocampal and rhinal areas (*Figure 2E* and *Figure 2—figure supplement 1D–E*).

In the intersection between prefrontal and temporal cortices, the UF coursed in close proximity to the insular cortex and regions in the caudal territory of the inferior frontal gyrus (*Figure 2A,E* and *Figure 2—figure supplement 1D–F*). More medially, human UF encompassed the lateral ventral striatum and kept a predominant lateral course in the ventral PFC (*Figure 2A* and *Figure 2—figure supplement 1D–F*), whereas a bundle of UF fibers stretched alongside the cingulate AmF and CB (*Figure 2C*). Here, the main body of the UF continued to run in a lateral position sending widespread innervation to all the ventrolateral PFC WM. Furthermore, a sub-section of fibers also coursed more medially, adjacent to sACC and more prominently to posterior OFC (*Figure 2A,E* and *Figure 2—figure supplement 1F*).

In the anterior territory of ventral PFC, the human UF lay along both medial and lateral orbital areas in a widespread fashion. A major bundle of fibers occupied a substantial part of the lateral OFC WM and adjacent cortical areas located on the ventral surface of the brain. Another lateral branch of the UF extended more rostrally in the WM flanking the lateral FPC, whereas medial FPC WM was primarily reached by the more medially coursing UF fibers (*Figure 2C,E* and *Figure 2—figure supplement 1F*).

Again, as for the AmF pathway, the anatomy of the UF as well as its interdigitation with AmF in the human brain was highly conserved in monkey and human brains (*Figure 2A–E* and *Figure 2—figure supplement 1D–F*). There may be differences in FPC organization between humans and macaques (*Neubert et al., 2015*) and it was notable that the UF innervated medial and lateral FPC WM in the human brain in equal proportion (*Figure 2E*).

## Comparison of the two fiber bundles

The main bodies of the human AmF and UF were anatomically segregated and constituted two anatomically distinct connectional systems in ATR. Alongside the dorsolateral amygdala, AmF interdigitated with more lateral temporo-frontal UF axons. At this level, it was possible to identify in both species a medial-lateral organization of the two bundles, with the AmF leaving the amygdalar WM more caudally and the UF fibers projecting to and from this subcortical structure in a more anterior position (*Figure 2*).

More anteriorly, the two tracts maintained a clear and evolutionarily preserved organization along a medial-lateral gradient, but also started to develop a dorsal-ventral relationship (*Figure 2A–E*). As previously described, in this region the AmF coursed medial to the ventral striatum, just underneath the anterior limb of the internal capsule and medial to the ventromedial striatum, specifically inferiorly to the internal and external globus pallidus, the ventral head of the caudate and the medial bank of the putamen (*Figure 2A–E* and *Figure 2—figure supplement 1A–B*). Conversely, the UF approached the basal ganglia more laterally. A medial portion of UF fibers interdigitated with lateral AmF axons and courses ventral to the AmF in the WM surrounding the putamen. In contrast to the more ventromedially positioned course, UF fibers run alongside the full ventrolateral extent of the basal ganglia. In the WM lateral to the putamen and ventral to the claustrum, UF fibers interlaced with the ventral axons of the EmC (*Figure 2A–E* and *Figure 2—figure supplement 1E–F*).

The relative mediolateral positioning of the AmF and UF extended more rostrally within the WM adjacent to the sACC (*Figure 2A* and *Figure 2D–E*). A major component of the AmF lay adjacent to sACC (*Figure 2—figure supplement 1C*), whereas only a medial bundle from the UF lay adjacent to sACC (*Figure 2A* and *Figure 2—figure supplement 1F*). The relative positioning of the two pathways then began to rotate partially, with the UF moving to a location ventral to the AmF as the anatomical distance between the two fiber bundles reduced in the subgenual WM. In the WM adjacent

to infralimbic cortex, the segregation of the two tracts began to fade as the dorsomedial UF fibers approached the ventrolateral AmF axons (*Figure 2A,F and G*).

Although the AmF and UF were clearly distinct entities in more posterior parts of the brain, their clear separation disappeared in the posterior OFC. Here, the UF and AmF merged into a single connectional system. This anatomical organization was present and conserved across both human and macaques in relation to several homologous landmarks in both species (*Figure 2B–E*; *Figure 2—figure supplement 1C and F*).

These results describe a preserved dichotomous organization of the AmF and UF along a predominantly medial-lateral axis in both species. As described in the introduction, the choice of these two fiber bundles was guided by tract-tracing results that suggested their connectivity of prefrontal regions with predominantly ATR fibers. As argued, tractography is not suitable for an investigation of full area-to-area connectivity, so instead we sought to investigate the contributions of connectivity with these gray matter areas by means of a connectivity-gradient analysis. Specifically, we were interested to see whether we could observe a similar medial-lateral gradient of connectivity within the fiber bundles, driven by amygdalar and anterior temporal polar connectivity. In different analyses, we quantified the relative probabilities that streamlines coursing through the core of the AmF and UF originating in the amygdala or in the temporal pole (see Materials and methods). The resultant connectivity-gradient ratio is represented in *Figure 3*, separately for AmF (left panel) and for UF (right panel). These analyses showed that a strong medial-lateral organization in the human brain is present not only across but also within tracts. This is particularly evident for the UF that contained streamlines originating from both the anterior temporal lobe and the amygdala, but with clear prevalence in the lateral and medial directions, respectively. A similar pattern is visible in the AmF, although it contained only very few streamlines originating in the temporal pole.

## Prefrontal connectivity and control tracts

We have described the distinct courses of the AmF and UF between the vicinity of the ATR and the prefrontal cortex. To quantify the streamlines of the tracts to prefrontal areas, we can describe their connectivity fingerprints with prefrontal regions of interest (*Mars et al., 2018*). This allows a quantitative comparison of the areal connectivity of AmF and UF. To examine the specificity of the results, we also compared the connectivity fingerprints of AmF and UF to those of three control tracts connecting nearby systems: the anterior limb of the internal capsule (ICa), the extreme capsule (EmC), and the cingulum bundle (CB) (*Figure 4*).

Comparing the connectivity fingerprints of AmF and UF, a medial-lateral organization in the density of streamlines between the two tracts was evident in both species (*Figure 4A–B*). Although both

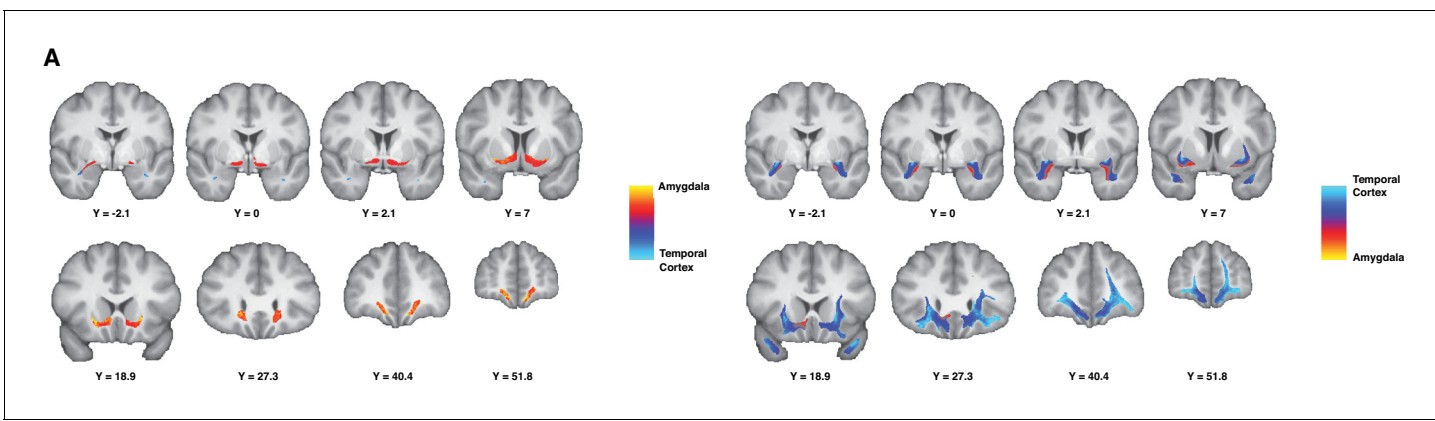

**Figure 3.** Gradient organization of ATR-prefrontal fibers. White matter gradients showing the organization of fibers coursing between ATR and prefrontal areas. In the left panel, we represent the connectivity gradient ratio of the AmF bundle, indexing the relative probability of AmF streamlines originating in either the amygdala (red-yellow) or the temporal cortex (blue). The color scale represents the full range of the gradient ratio, from solely originating in the amygdala (yellow), to solely originating in the temporal cortex (light blue). In the right panel, we represent the same gradient ratio for the UF bundle.
DOI: https://doi.org/10.7554/eLife.47175.005

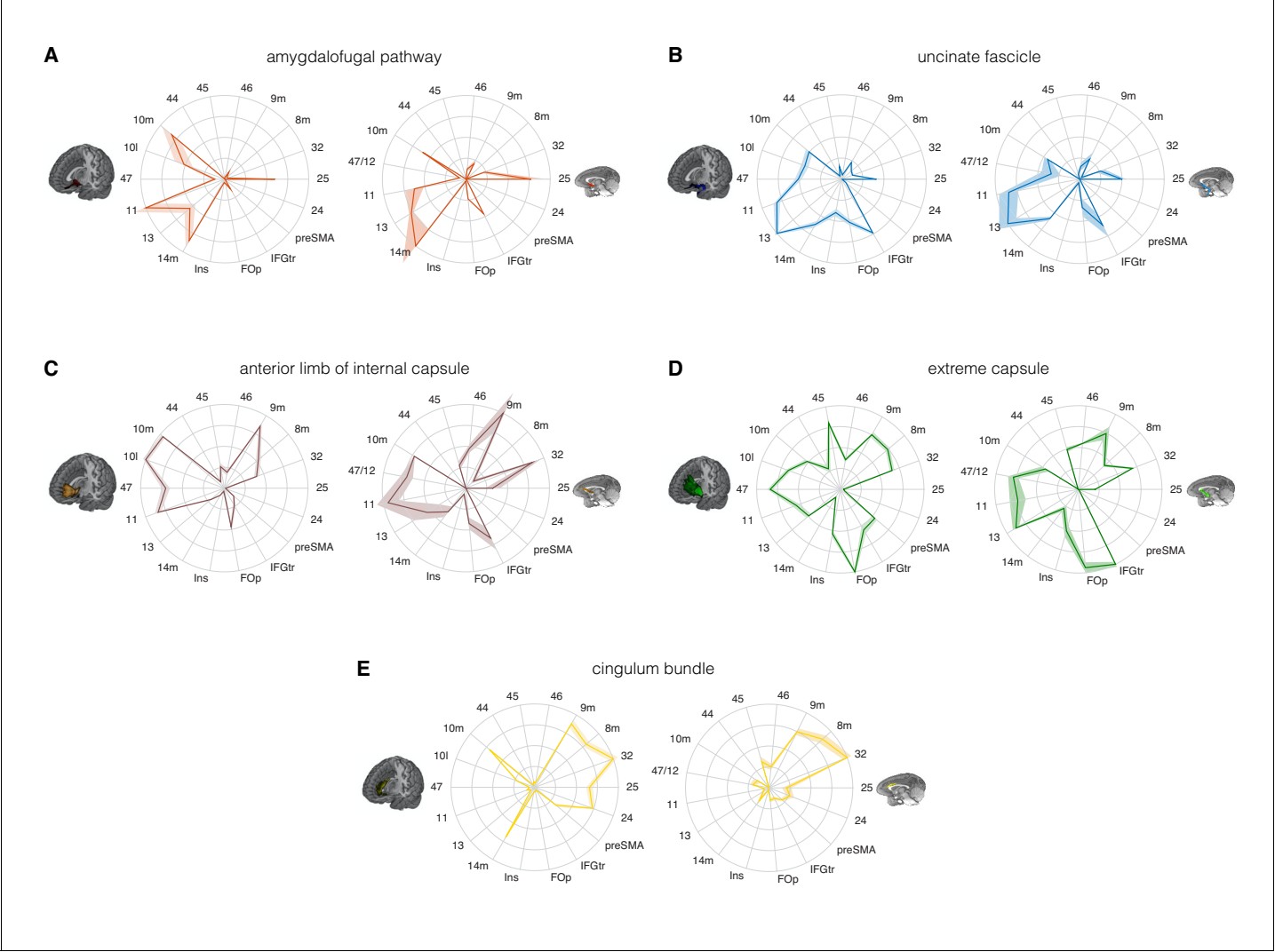

**Figure 4.** Comparison of the connectivity fingerprints of the amygdalofugal pathway, uncinate fascicle and adjacent white matter tracts in the human (left) and macaque (right) frontal lobe. (A) The pattern of AmF fibers is preserved across the two species and predominantly targets medial areas in subgenual, orbital, ventral, and polar PFC. (B) Overall, UF axons show high similarity between the two species, but stronger innervation to the opercular and insular cortex is observable in the human UF. (C–E) Macaque–human comparison of the prefrontal fibers organization of three connectional systems located in the white matter dorsal to the AmF (internal capsule; C), dorsal to the UF (extreme capsule; D) or interacting with both tracts in the infralimbic and prelimbic cortex (cingulum bundle; E). 9 m, dorsomedial PFC; 8 m, caudal bank of the dorsomedial PFC; 32, anterior cingulate cortex; 25, subgenual cingulate cortex; 24, dorsal anterior cingulate cortex; preSMA, pre-supplementary motor area; IFGtr, inferior frontal gyrus, pars triangularis; FOp, frontal operculum; Ins, insular cortex; 14 m, ventromedial PFC; 13, caudal orbitofrontal cortex; 11, rostral orbitofrontal cortex; 47 (human) or 47/12 (macaque) lateral orbitofrontal cortex (also named lateral convexity in *Murray and Rudebeck, 2018* and in *Rudebeck and Murray, 2014*); 10 l, lateral frontopolar cortex; 10 m, medial frontopolar cortex; 44, caudal bank of the ventrolateral PFC; 45, rostral bank of the ventrolateral PFC. Shades represent standard error of the mean (SEM).

DOI: https://doi.org/10.7554/eLife.47175.006

The following figure supplements are available for figure 4:

**Figure supplement 1.** Methodological pipeline and white matter seed masks used to reconstruct each pathway.
DOI: https://doi.org/10.7554/eLife.47175.007

**Figure supplement 2.** Target regions of interest used to estimate the prefrontal connectivity fingerprint for each tract.
DOI: https://doi.org/10.7554/eLife.47175.008

bundles primarily reached ventral prefrontal cortex in both species, AmF was more likely to reach medial areas such as 25, 14 m, 11 and 10 m (*Figure 4A*), whereas UF primarily reached more lateral areas such as the inferior frontal gyrus, insular cortex, frontal operculum, area 47 (or the macaque

homolog 47/12), and area 45 (*Figure 4B*). However, UF fibers in the rostral ventral PFC were also in contact with more medial OFC (area 11) and FPC (10 m). Some species-specific differences were also evident. Human AmF fibers were denser in areas 11 and 10 m, whereas macaque AmF showed this pattern in area 32 and caudal vmPFC area 14 m (*Figure 4A*). Inter-species differences in cingulate fibers may partly be the result of species-specific idiosyncrasies in neuroanatomical geometries, as can be seen especially when diffusion MRI is used to reconstruct tracts running in the WM adjacent to brain structures presenting different degrees of curvatures in different species. UF also exhibited some contrasting inter-species differences. In the human brain, the numbers of streamlines reaching lateral orbital, ventrolateral, insular, and opercular areas were higher than the corresponding numbers in the macaque brain, suggesting a more widespread pattern of connectivity in human prefrontal cortex (*Figure 4B*). This may reflect prefrontal cortical expansion in humans compared to that in other Old World primates (*Smaers et al., 2017*). As discussed above, in the most anterior part of the prefrontal cortex, AmF and UF tend to intermingle, and it is therefore not surprising to see that both reached the frontal pole (area 10 m), including the human lateral frontal pole (area 10 l) for which the homology in the macaque monkey brain is debated (*Neubert et al., 2015*).

The connectivity fingerprints of the control tracts were markedly distinct from those of both AmF and UF (*Figure 4C–E*). The anterior limb of the internal capsule carries fibers originating and terminating in the thalamus and brainstem (*Lehman et al., 2011*). Frontally, its streamlines predominantly reached more dorsomedial regions than AmF and UF, including areas 32, 8 m, and 9 m (*Figure 4C*). The connectivity pattern of EmC was even more widespread in relation to areas of lateral and dorsal prefrontal cortex, including areas 45, 8 m, 9 m, and 32 (*Figure 4D*). The cingulum bundle showed high streamline count in medial areas (*Figure 4E*). Interestingly, similarly to the UF, some branches of these control tracts exhibited a more widespread pattern of connectivity in the human brain compared to the macaque brain. Indeed, human EmC streamline counts were higher in more caudal areas of dorsal PFC such as the preSMA (*Figure 4D*). CB fibers, instead, showed a trend in frontopolar, orbital, subgenual and midcingulate cortex that was similar to that in the macaque (*Figure 4E*).

The connectivity fingerprints can be used to compare the connectivity profiles across species quantitatively. We can create a dissimilarity matrix between species by calculating the Manhattan distance for each possible pair of tracts. This measure indicates how different two connectivity fingerprints are: the best matching pairs are defined as the set of fingerprints that have the smallest distance values (*Mars et al., 2016b*). When comparing the fingerprints (averaged across the two hemispheres) between species, we saw that the smallest differences for each tract were on the diagonal of the distance matrix (*Figure 5B*), indicating that each tract had a connectivity fingerprint most similar to its counterpart in the other species. AmF and UF are each other's second best choice, but matched best with their counterpart. This similarity is not unexpected given the close proximity in ventral PFC and considering the strong interconnection and reciprocal horizontal projections of the OFC regions. Yet despite these correspondences, the dissimilarity matrix confirms that the two pathways are distinguishable.

These results suggest that the AmF and UF streamlines reach different parts of prefrontal cortex and, moreover, that their connectivity profiles are distinct from those of nearby fiber bundles. This dissimilarity with respect to the control tracts was also reflected in their course through the white matter (*Figure 5A*). To provide a more complete picture of the organization of AmF and UF in the human brain, a clear topographic organization of these two fiber bundles relative to the three tracts described above in the human brain was observed. Indeed, the course of both tracts could be dissociated from ICa, EmC and CB but also with respect to two other tracts, namely the stria terminalis (ST) and fornix (Fx), that lie and course in temporal portions of WM near to AmF and UF (*Figure 5A*). Because prefrontally ST and Fx primarily innervate basal forebrain and limbic regions and have poor or absent projections to more rostral PFC areas (*Aggleton, 2008*; *Catani et al., 2002*; *Kitt et al., 1987*; *Oler et al., 2017*), with only some ST fibers reaching the posterior vmPFC (*Lehman et al., 2011*), these tracts were not included in the dissimilarity matrix analysis (*Figure 5B*).

## Discussion

In this study we use diffusion MRI tractography in primates **to** identify and define two closely related but distinct connectional systems linking ATR and PFC. The structure, course, and projections of

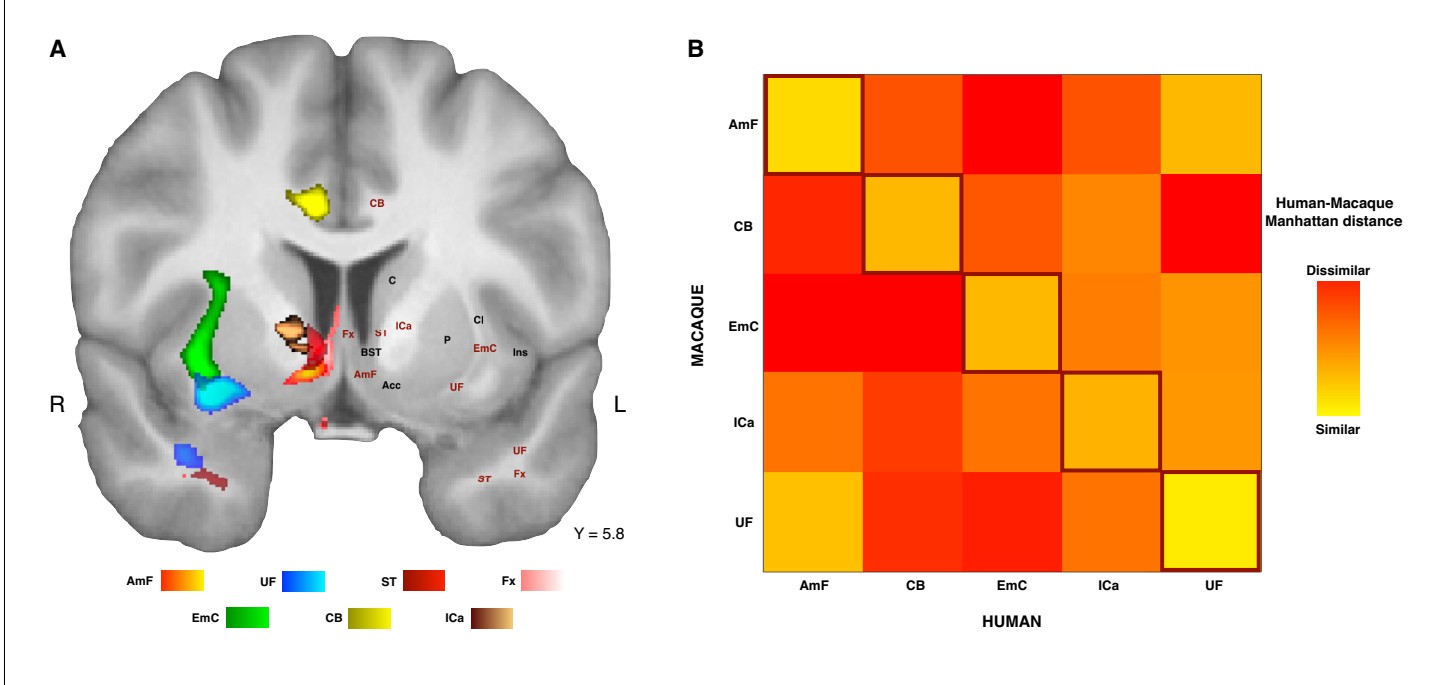

**Figure 5.** Organization of ATR-prefrontal tracts. (**A**) Topographic organization of the AmF and UF with respect to other urrounding WM tracts. Despite the close proximity of these five bundles, they occupy different portions of white matter and can be dissociated from one another. (**B**) Dissimilarity matrix of the prefrontal course of the tracts of interest across species. The matrix shows a quantitative comparison of the connectivity fingerprints of the five frontal tracts that were reconstructed in both the human and the monkey brains. Human tracts are listed on the horizontal axis; macaque tracts are listed on the vertical axis. Pair-wise dissimilarity is defined as the 'Manhattan distance' between each pair of pathways across the two species. The greatest similarity was found when comparing each tract with its homolog in the other species (bright diagonal in the dissimilarity matrix). AmF, ventral amygdalofugal pathway; CB, cingulum bundle; ICa, anterior limb of the internal capsule; EmC, extreme capsule; UF, uncinate fascicle; ST, Stria Terminalis; Fx, Fornix.

DOI: https://doi.org/10.7554/eLife.47175.009

The following figure supplement is available for figure 5:

**Figure supplement 1.** White matter seed masks for two limbic control tracts in the human brain.

DOI: https://doi.org/10.7554/eLife.47175.010

these tracts in the macaque monkey closely match those observed in previous tracer- injection studies (*Fudge et al., 2002*; *Fudge and Haber, 2001*; *Kunishio and Haber, 1994*; *Lehman et al., 2011*). Moreover, we showed that the connectional profiles of these two tracts are preserved across humans and monkeys. We demonstrated that both the AmF and UF run separately and in parallel when exiting the ATR but then merge to form a single fiber complex in the ventral PFC.

Tract tracing studies provide important information about where specific axons in fiber bundles terminate or originate in the brain (*Petrides and Pandya, 2007*). However, the technique is invasive and requires the sacrifice of the subject, limiting its use in human anatomy. Diffusion MRI tractography excels when tracing in homogenous white matter and can reconstruct many of the major white matter tracts. However, it can fail in areas in which fiber density is low or where different fibers cross. These properties are exhibited by the porous heterogenous tissue in the basal forebrain that the amygdalofugal pathway traces through, as evidenced by invasive tracer injection and histology studies. It would be especially difficult to track in these circumstances using algorithms that do not utilize the full probability distribution of the fiber directions. Perhaps because of these difficulties, previous studies using tractography to investigate the limbic system might have overlooked or undervalued the prefrontal projections of the AmF (*Catani et al., 2013*; *Croxson et al., 2005*; *Kamali et al., 2016*). To overcome these challenges, we employed a strategy of first performing probabilistic tractography based on crossing-fiber resolved diffusion models in the macaque monkey to develop a protocol that was able to reconstruct the AmF fibers as known from previous invasive tracer studies in that species (*Oler et al., 2017*). This strategy was successful. We were able to reconstruct the

course of the tract and confirmed the AmF as primarily a subcortical limbic tract, but with prefrontal fibers mostly extending along medial areas such as 25 and 14 m. This pattern was similar in the human brain.

The division between AmF and UF bundles maps onto the tracer-based division of the ventral PFC into a medial region and a more lateral/orbital region, as originally suggested by Price and colleagues (*Carmichael and Price, 1995*; *Ongür et al., 2003*). Following the logic that the connections of an area are a crucial determinant of its function, this suggested a different functional role of these subdivisions with the medial network strongly connected with visceromotor areas and the lateral orbital network interconnected with multi-modal sensory inputs (*Ongür and Price, 2000*). This was confirmed by dissociations apparent from lesion studies with medial frontal lesions showing an impairment in the capacity to maintain sustained arousal when anticipating rewards (*Rudebeck et al., 2014*), but temporal and ventral lateral frontal lesions leading to an impairment in learning arbitrary associations between stimuli and outcomes or stimuli and actions (*Browning and Gaffan, 2008*; *Eacott and Gaffan, 1992*; *Gutnikov et al., 1997*). Our results show divergences across AmF and UF in ventral PFC that supports this dissociation and demonstrate that it generalizes to the human brain. Fibers in AmF and UF may constitute the primary white matter scaffolding from which these two networks may arise.

The importance of understanding a dissociation between AmF and UF may be further underlined by considering the directionality of influence of these two bundles as described by more invasive investigations: the UF is a bidirectional tract with many projections from prefrontal cortex terminating in medial and lateral basal nuclei of the amygdala, whereas the AmF predominantly carries efferent projections from the amygdala to subcortical and prefrontal targets (*Bickart et al., 2012*). Potentially, this dissociation can help us to understand whether abnormal neural processes that are central to certain psychiatric disorders are more related to efferent or afferent projections of the amygdala (*Bramson et al., 2019*; *Murray and Rudebeck, 2018*; *Volman et al., 2013*). Unfortunately, one of the limitations of probabilistic tractography is that it cannot infer the directionality of the reconstructed bundles and is unable to differentiate monosynaptic from polysynaptic connections. For this reason, the combination of tracer and imaging studies to translate results from macaque to human brains is an essential approach that allows us to better understand human neuroanatomy in a non-invasive fashion.

It is important to note, however, that the dissociation between the medial and lateral circuits is not complete, with the circuits joining up in anterior OFC and rostral prefrontal cortex. An integration between the two processing streams has been suggested both on anatomical and functional grounds. Anatomically, the more rostral parts of the prefrontal cortex are suggested to be more interconnected (*Carmichael and Price, 1995*). Functionally, such an integration would subserve learning and decision making computations by allowing the integration of learned stimulus-outcome contingencies with the internal state of the organism. This is evident in the proposal of a 'common currency' representation comprising the value of different objects irrespective of the physical properties (*Montague and Berns, 2002*; *Padoa-Schioppa and Assad, 2008*). The rostral interactions of AmF and UF in areas 11, 13 and 14 m may therefore represent anatomical support for such computations.

It is important to point out that our use of diffusion-weighted MRI tractography necessitated some different protocol choices than would be made for a conventional tract tracing study. A number of recent studies have criticized tractography for its perceived lack of reliability in areal-to-areal connections (*Reveley et al., 2017*; *Donahue et al., 2016*). Therefore, rather than replicating the tracer approach of seeding in circumscribed areas and assessing which gray matter regions are reached, we choose to seed in the core of the white matter bundles. Here, the tractography algorithm is less impaired and not affected by gyral bias or superficial white matter. Only in follow-up analyses did we investigate whether tractography seeded in these bundles reached gray matter targets or whether gray-matter-seeded tractography reaches a certain bundle. By placing the bundle at the core of the analyses, our results become more reliable. This approach also allows us to create more easily tractography recipes for the fiber bundles of interest that can then be shared with other researchers (e.g., *Bramson et al., 2019*).

The importance of considering neural circuitry, extending beyond individual brain regions, is increasingly evident when studying the contributions of ventral PFC in emotion, learning, and decision making. Combining controlled focal lesion and transient perturbation studies with whole-

brain neuroimaging highlights the critical role of remote connections to the impact of local perturbations (*Folloni et al., 2019*; *Froudist-Walsh et al., 2018*; *Verhagen et al., 2019*). Moreover, it has recently been demonstrated that some lesion effects might be better explained by damage to connections as opposed to damage to single regions (*Rudebeck et al., 2013*; *Thiebaut de Schotten and Foulon, 2018*), and differences in connectivity have already been argued to play a crucial role in regional specialization in other brain areas (*Krubitzer, 2007*). Given the inverse problem in which changes in brain activity can have multiple causes (*O'Reilly et al., 2013*), an understanding of connectional anatomy is essential to understand these network effects. As such, this work can help us to move from a view of ventral PFC as a constellation of isolated, 'functionally specialized' regions to a more network-based approach, thereby helping us to describe the underlying neurobiology of psychiatric disorders more appropriately and to promote treatment options more effectively.

Conventionally, surgical interventions, including deep brain stimulation for depression, addiction, and obsessive-compulsive disorders were primarily targeted at gray matter tissue. For instance, aspects of area 25 anatomy are associated with the presence of psychiatric disorders (*Drevets et al., 1997*; *Murray et al., 2011*). Indeed, chronic deep brain stimulation in this area improves chronic dysphoric conditions (*Mayberg et al., 2005*). However, with evidence accumulating that the most effective electrodes in deep brain stimulation are situated in or close to white matter fiber tracts, rather than in gray matter proper, recent deep brain stimulation approaches are explicitly targeting white matter fiber structures (*Baldermann et al., 2019*; *Lozano et al., 2019*). Area 25 is innervated and visited by the nearby AmF, suggesting the hypothesis that the physiological effects of effective electrical stimulation of this region (*Johansen-Berg et al., 2008*) may be ascribed to the AmF pathway and the regions it interconnects.

For this study, we used high- quality diffusion-weighted MRI data, obtained by scanning ex-vivo macaque samples for multiple hours, and by using in-vivo human diffusion MRI data with an acquisition duration of one hour. Both of these datasets relied on unconventionally strong diffusion gradients, high number of diffusion directions, and high spatial resolution. However, the tractography recipes developed as part of this study have already been shown to be applicable and successful in human diffusion MRI data of more conventional quality. In fact, the approach and results achieved here, including the division between these two amygdala-prefrontal tracts, are behaviorally relevant in the context of social-emotional actions, advancing our understanding of the neurobiology of emotion regulation (*Bramson et al., 2019*).

To conclude, two principles underlie the organization of amygdala–PFC connectivity in the human brain. They are both evolutionarily conserved and shared with the macaque monkey. First a 'connectional segregation principle' means that AmF and UF occupy a relative mediolateral position with respect to one another. Second, a 'connectional integration principle' means that the tracts merge together as they enter and progress through the WM in the PFC to create an intricate network of projections from both cortical and subcortical regions. The two connectional systems described here may support distinct but synergic brain functions in PFC and this hypothesis should be investigated in future to understand how regions in the temporal lobe and PFC interact during affective, social, and cognitive tasks, how they could be disturbed in psychiatric disorders, and how this can be alleviated using targeted brain-stimulation approaches.

## Materials and methods

### Data acquisition

Human in-vivo diffusion MRI data were provided by the Human Connectome Project (HCP), WU-Minn Consortium (Principal Investigators: David Van Essen and Kamil Ugurbil; 1U54MH091657) funded by the 16 NIH Institutes and Centers that support the NIH Blueprint for Neuroscience Research; and by the McDonnell Center for Systems Neuroscience at Washington University (*Van Essen et al., 2013*). The minimally preprocessed datasets of the first 24 subjects from the Q2 public data release were used (age range 22–35 years; 13 females). No data were excluded from the analysis. This study did not include an experimental manipulation. Accordingly, there was no pre-selection nor restriction for group allocation. Data acquisition protocols are detailed in *Uğurbil et al. (2013)* and *Sotiropoulos et al. (2013)*. The diffusion data were collected at a 1.25 mm isotropic resolution across the entire brain on a customized 3T Siemens Skyra scanner using a monopolar

Stejskal-Tanner diffusion scheme with a slice-accelerated EPI readout. Sampling in q-space included three shells at b = 1000, 2000, and 3000 s/mm$^2$. For each shell, 90 diffusion gradient directions and six non-diffusion weighted images (b = 0 s/mm$^2$) were acquired with reversed phase-encoding direction for TOPUP distortion correction (*Andersson et al., 2003*). A subject-specific cortical surface mesh with 10 k vertices was created using FreeSurfer on the T1-weighted image (acquired using an MPRAGE sequence at 0.7 mm isotropic resolutions) and aligned to the diffusion space as part of the HCP's minimum preprocessing pipeline (*Glasser et al., 2013*).

Macaque diffusion MRI data were obtained ex-vivo from three rhesus monkeys (*Macaca mulatta*, age range at death 4–14 years, mean age 8 years, standard deviation 6.6; one female) using a 7T magnet with a Varian DirectDrive (Agilent Technologies, Santa Clara, CA, USA). All intact post-mortem brain samples available at the time of the investigation were included in this study. No data were excluded from the analysis. Immediately after death, the brains were perfusion fixed with formalin and stored. Approximately one week before MRI scanning, the brains were perfused in phosphate buffer solution to enhance their diffusion signal. During scanning, the brains were suspended in fomblin, which has no magnetic properties itself and which therefore allows an unbiased scan. For each brain, nine non-diffusion-weighted (b = 0 s/mm$^2$) datasets and a single diffusion-weighted (b = 4000 s/mm$^2$) dataset were acquired using a single line readout, 2D Stejskal-Tanner pulse sequence (*Stejskal and Tanner, 1965*) (TE/TR 25 ms/10 s; matrix size 128 × 128; resolution 0.6 mm x 0.6 mm; 128 slices; slice thickness 0.6 mm; 131 isotopically distributed diffusion directions).

In addition, in the same session for each brain, a T1-map, resulting in a T1-weighted image, was collected (MPRAGE sequence at TE/TR 8 ms/10 s; Ti 10.6 ms; matrix size 128×128; 128 slices; slice thickness 0.6 mm). Death and fixation have an impact on the brain, causing a reduction in ex-vivo tissue diffusivity. Therefore, to achieve an equivalent diffusion contrast to that of in-vivo data, the diffusion coefficient was increased from b = 1000 to 4000 s/mm$^2$ (*Mars et al., 2016a*) Importantly, although diffusion magnitude is affected and requires compensation, diffusion anisotropy is largely preserved post mortem (*D'Arceuil and de Crespigny, 2007*).

## Data preprocessing

Human data were preprocessed according to the HCP minimal preprocessing pipelines (*Glasser et al., 2013*; *Sotiropoulos et al., 2013*). Macaque data preprocessing and all subsequent analyses were performed using tools from FSL (*Smith et al., 2004*) and the in-house MR Comparative Anatomy Toolbox (MrCat), and MRIcroGL (www.mricro.com). In short, non-diffusion-weighted images were extracted from the full set, averaged, and corrected for spatial RF-field inhomogeneity bias before being linearly registered to the average T1-weighted structural image of the same animal. The T1-weighted images in turn were linearly and non-linearly registered to a dedicated ex-vivo macaque T1w template aligned to the standard macaque F99 space (*Van Essen and Dierker, 2007*). Combined, these registrations allowed a direct mapping between diffusion-weighted, T1-weighted, and standard space for both human and macaque data.

Diffusion tensors were fitted to each voxel using DTIFIT (*Behrens et al., 2003*). Voxel-wise crossing-fiber model fitting of diffusion orientations was performed for both human and macaque datasets using FSL's BedpostX (*Behrens et al., 2007*). For the human in-vivo data, a multi-shell extension was used to reduce overfitting of crossing fibers due to non-monoexponential diffusion decay (*Jbabdi et al., 2012*). Up to three fiber orientations per voxel were allowed. This produced voxel-wise posterior distributions of fiber orientations that were subsequently used in probabilistic tractography.

## Probabilistic tractography

The principal aim of the tractography analyses was to reconstruct the prefrontal projections of the ventral amygdalofugal pathway (AmF) and uncinate fascicle (UF). Tractography recipes for both were created by reference to previously published atlases (*Nieuwenhuys et al., 2008*; *Paxinos et al., 2000*), tract-tracing studies (*Fudge et al., 2002*; *Lehman et al., 2011*; *Schmahmann and Pandya, 2006*), and diffusion tractography studies (*Catani et al., 2013*; *Croxson et al., 2005*; *Jbabdi et al., 2013*). All masks were created in F99 space (*Van Essen and Dierker, 2007*) in the monkey brain and MNI space in the human brain, and subsequently warped to each individual's diffusion space and adjusted by hand to ensure anatomical accuracy.

The seed mask of the AmF was drawn in the sub-commisural white matter perforating the substantia innominata, a region between the dorsal amygdaloid and the bed nuclei of the stria terminalis (BNST). The mask was carefully aimed to reproduce macaque tract tracing studies (*deCampo and Fudge, 2013*; *Fudge and Haber, 2001*; *Schmahmann and Pandya, 2006*), located medially to the ventral pallidum (VP) and sublenticular extended amygdala (SLEA) and dorsally to the nucleus basalis of Meynert (NBM) (*Mai et al., 2015*; *Paxinos et al., 2000*). In accordance with the local trajectory of the AmF (as estimated by tract tracing), the seed mask was constrained to contiguous voxels showing high fractional anisotropy in an anterior-posterior direction (*Figure 4—figure supplement 1B*). UF was seeded axially in the anterior temporal lobe, in the white matter rostro-lateral to the amygdala (*Figure 4—figure supplement 1C*). After testing various coronal seed masks, an axial seed mask was chosen to account for the strong curve of the fibers from dorsal-ventral to anterior-posterior orientation to the orientation as they enter into the frontal lobe (*Catani and Thiebaut de Schotten, 2008*).

To test the specificity of our tractography results, we reconstructed three alternative tracts projecting to the prefrontal cortex: the anterior limb of the internal capsule (ICa), the extreme capsule (EmC), and the cingulum bundle (CB). These tracts were chosen because together they constitute prominent systems connecting thalamic, anterior temporal, and limbic regions with PFC regions, and because they are expected to run or terminate in close proximity to UF and AmF. ICa was seeded in the ventral section of the anterior limb of the internal capsule (*Figure 4—figure supplement 1D*) passing between the caudate and pallidum, superior to the anterior commissure, and inferior and lateral to the caudate, in a position similar to that used by *Jbabdi et al. (2013)*. The EmC seed mask was drawn in the sheet of white matter between the putamen and insula (*Figure 4—figure supplement 1E*). The seed voxels were placed as laterally as possible within this sheet to favor fibers of the extreme over the external capsule, although due to the resolution of diffusion imaging, it is not possible to ensure fully that no fibers of external capsule were included in the reconstructed EmC (*Mars et al., 2016a*). EmC seed voxels were placed just superior to the UF, in order to avoid including fibers belonging to this latter tract. The CB seed mask was drawn in a coronal plane capturing the WM dorsal and medial to the corpus callosum, within the cingulate gyrus (*Figure 4—figure supplement 1F*). Two additional control tracts, namely the stria terminalis (ST) and fornix (Fx), were reconstructed in the human brain to address their differentiation from AmF and UF. The ST seed mask was placed in its caudal part, just anterior to the curve that this bundle makes in the proximity of the splenium of the corpus callosum (*Figure 5—figure supplement 1A*). The Fx seed mask was drawn at the same level as the ST seed mask but in a more medial position, in a portion of WM just ventral to the corpus callosum (*Figure 5—figure supplement 1B*).

To constrain the tractography algorithm, exclusion masks were drawn in both species to exclude false- positive results in areas of high crossing fibers as follows: (1) within the basal ganglia to avoid picking up spurious subcortical tracts (except for ICa tracking); (2) posteriorly to the seeds to prevent the projections from running backwards as the prefrontal cortical streamlines were the focus of this study; (3) an axial slice at the level of the superior temporal gyrus to prevent tracts from running in a ventral direction in an unconstrained manner (except for UF, CB, and AmF); (4) on the axial-coronal slices cutting across the thalamus, basal ganglia, and corpus callosum to exclude these subcortical and callosal projections; (5) in the dorsal cingulate cortex to avoid leakage of tracts to nearby bundles as a result of high curvature of the tracts (except for CB tracking); (6) an L-shaped coronal mask from the paracingulate cortex and the inferior frontal sulcus to the vertex to exclude tracking in the superior longitudinal fascicle; (7) another L-shaped mask, only for ST and Fx, drawn at the level of the retrosplenial cortex and dorsal to the corpus callosum to exclude inappropriate leakage into dorsal parietal and dorsal frontal cortices; and (8) the opposite hemisphere to only track ipsilateral tracts. An exclusion mask of the CSF was used in each subject to prevent fibers from jumping across sulci during tracking. Streamlines encountering any of the exclusion masks were excluded from the tractography results. A coronal waypoint section was drawn in the frontal lobe at the level of the caudal genu of the corpus callosum to ensure that the fibers emanating from the seeds were projecting to the prefrontal cortex; this prevented leakage of the EmC into the corticospinal tract. For the ST and Fx only, two tract-specific coronal masks were placed in the temporal and forebrain white matter.

Probabilistic tractography was performed in each individual's diffusion space. All seeding and tracking parameters were kept constant across tracts and only adjusted between species to deal

with differences in brain size and data resolution. Human tractography parameters were: maximum of 3200 steps per sample; 10,000 samples; step size of 0.25 mm; and curvature threshold of 0.2. Macaque tractography parameters were: maximum 3200 steps per sample; 10,000 samples; step size of 0.1 mm; and curvature threshold of 0.2.

A visitation map or 'tractogram' was constructed for each individual in order to allow comparison of these maps between tracts, subjects, and species. Each tractogram was log-transformed to account for the exponential decrease of visitation probability with distance and normalized by dividing each voxel's value by the 75th percentile value across the tractogram, thereby removing potential bias of differences in numbers of streamlines between tracts and across species. In both human and macaques, the focus of the investigation was on the organization of the tracts in prefrontal cortex. Prefrontal streamlines of each tract were averaged in a species-specific group template tractogram. As such, the center of the group is defined as the mean of the individual subject tractograms. For the purposes of visualization only, the normalized tractograms were subsequently thresholded with minimum and maximum values equal to 0.5 and 2, respectively.

We further asked how streamlines origination in either the amygdala or temporal pole were organized with respect to each other within the AmF and UF bundles. To this end, we performed a within-bundle connectivity-gradient analysis. We placed seeds in the amygdala (Amyg) and the temporal pole (TP) and used the core of the AmF and UF tracts as waypoints (i.e. these waypoint masks were identical to the seed masks of the previous analyses). This resulted in four normalized tractograms, masked by the bundles as described above: $AmF_{Amyg}$, $AmF_{TP}$, $UF_{Amyg}$, and $UF_{TP}$. A gradient ratio describing the relative occurrence of Amyg and TP streamlines was defined for the AmF bundle as: $ratio_{AmF} = \frac{AmF_{Amyg} - AmF_{TP}}{AmF_{Amyg} + AmF_{TP}}$, and for the UF bundle as: $ratio_{UF} = \frac{UF_{Amyg} - UF_{TP}}{UF_{Amyg} + UF_{TP}}$. In other words, these indices quantify the relative probability that streamlines coursing through the AmF and UF bundles have originated from either the amygdala (positive values) or the temporal pole (negative values).

## Connectivity fingerprints

The differential distribution of streamlines of the reconstructed tracts within areas in prefrontal cortex were quantified by creating so-called connectivity fingerprints (*Mars et al., 2018*; *Passingham et al., 2002*) of each tract with regions of interest (ROIs) in prefrontal cortex. Such fingerprints can be used to compare tracts' prefrontal projections with one another and to compare the tracts across species. Human and macaque prefrontal ROIs were created on the basis of previously published coordinates (*Neubert et al., 2015*; *Neubert et al., 2014*; *Sallet et al., 2013*) or drawn by hand on the basis of anatomical atlases (*Mai et al., 2015*; *Paxinos et al., 2000*; *Petrides et al., 2012*; *Yeterian et al., 2012*) (*Figure 4—figure supplement 2*). Each coordinate was projected onto a subject-specific cortical surface mesh with 10,000 vertices (10 k) in the human and to a template 10 k surface in the macaque. Here, on the surface representation, these coordinates were expanded to circular ROIs following the cortical contours, with a 10 mm radius in humans and a 3 mm radius in macaques. The resulting ROIs were then projected back to each subject's native brain image in volumetric space. The creation of ROIs on the surface rather than in volumetric space allows us to account for individual idiosyncrasies in the anatomy of sulci and gyri, ensuring that we have truly cortical rather than spherical ROIs. The white matter/gray matter border of each ROI was then extracted and used as the target region in the connectivity fingerprint. This approach allows us to maximize the estimation of streamlines projecting to each region while minimizing the reduction of signal consequent to the tissue-related poor anisotropic signal characteristic of ROIs created in gray matter only. The fingerprints are then constructed by counting, for each tract, the average number of streamlines hitting the voxels of each ROI. To allow comparison across tracts and species, these values were normalized by dividing each tract-fingerprint by its maximum values across the selected ROIs. The center of the group is defined as the mean of the individual subject connectivity fingerprints. Connectivity fingerprints can then be compared by calculating the Manhattan distance between them (*Mars et al., 2016b*).

## Acknowledgements

We would like to thank Matthew FS Rushworth and Richard E Passingham for valuable discussions on the results. The work of DF is supported by the Wellcome Trust [UK Grant 105238/Z/14/Z] and by

the Medical Research Council [CQR01330]. The work of JS is supported by a Wellcome Trust Henry Dale Fellowship (105651/Z/14/Z). The work of LV is supported by the Wellcome Trust [WT100973AIA] and by a Marie Curie Intra- European Fellowship within the European Union's 7th Framework Programme [MC-IEF-623513]. The work of RBM is supported by the Biotechnology and Biological Sciences Research Council (BBSRC) UK [BB/N019814/1] and by the Netherlands Organization for Scientific Research NWO [452-13-015]. The Wellcome Centre for Integrative Neuroimaging is supported by core funding from the Wellcome Trust [203139/Z/16/Z]. NRS and AAK were supported by Cancer Research UK [grant C5255/A15935].

## Additional information

### Funding

| Funder | Grant reference number | Author |
|---|---|---|
| Wellcome | 105238/Z/14/Z | Davide Folloni |
| Wellcome | 105651/Z/14/Z | Jerome Sallet |
| Wellcome | WT100973AIA | Lennart Verhagen |
| European Commission | MC-IEF-623513 | Lennart Verhagen |
| Medical Research Council | CQR01330 | Davide Folloni |
| Wellcome | 203139/Z/16/Z | Davide Folloni<br>Jerome Sallet<br>Lennart Verhagen<br>Rogier B Mars |
| Biotechnology and Biological Sciences Research Council | BB/N019814/1 | Rogier B Mars |
| Netherlands Organisation for Scientific Research | 452-13-015 | Rogier B Mars |
| Cancer Research UK | C5255/A15935 | Alexandre A Khrapitchev<br>Nicola Sibson |

The funders had no role in study design, data collection and interpretation, or the decision to submit the work for publication.

### Author contributions

Davide Folloni, Conceptualization, Resources, Data curation, Software, Formal analysis, Funding acquisition, Validation, Investigation, Visualization, Methodology, Writing—original draft, Project administration, Writing—review and editing; Jerome Sallet, Conceptualization, Resources, Data curation, Formal analysis, Supervision, Funding acquisition, Validation, Investigation, Visualization, Methodology, Project administration, Writing—review and editing; Alexandre A Khrapitchev, Data curation, Methodology, Writing—review and editing; Nicola Sibson, Data curation, Funding acquisition, Methodology; Lennart Verhagen, Rogier B Mars, Conceptualization, Resources, Data curation, Software, Formal analysis, Supervision, Funding acquisition, Validation, Investigation, Visualization, Methodology, Writing—original draft, Project administration, Writing—review and editing

### Author ORCIDs

Davide Folloni https://orcid.org/0000-0003-1653-5969
Jerome Sallet http://orcid.org/0000-0002-7878-0209
Alexandre A Khrapitchev http://orcid.org/0000-0002-7616-6635
Lennart Verhagen https://orcid.org/0000-0003-3207-7929
Rogier B Mars https://orcid.org/0000-0001-6302-8631

### Ethics

Human subjects: Human in-vivo diffusion MRI data were provided by the Human Connectome Project (HCP), WU-Minn Consortium (Principal Investigators: David Van Essen and Kamil Ugurbil;

1U54MH091657) funded by the 16 NIH Institutes and Centers that support the NIH Blueprint for Neuroscience Research; and by the McDonnell Center for Systems Neuroscience at Washington University (Van Essen et al., 2013). The ex-vivo non-human primate samples used in this work were made available from an existing post-mortem collection at the University of Oxford. This work did neither impact, nor involve decisions on the health, life, or death of any animal. As such, the research does not require a Home Office License as defined by the Animals (Scientific Procedures) Act 1986 of the United Kingdom. Accordingly, the use of post-mortem samples for this work does not require additional ethical approval.

## Decision letter and Author response

Decision letter https://doi.org/10.7554/eLife.47175.016
Author response https://doi.org/10.7554/eLife.47175.017

## Additional files

### Supplementary files

• Transparent reporting form DOI: https://doi.org/10.7554/eLife.47175.011

### Data availability

The tractograms created as part of this study will be made available from the lab's website (www. neuroecologylab.org) and the Wellcome Centre for Integrative Neuroimaging Git (https://git.fmrib. ox.ac.uk/davidef/amygtemp_pfc), as will tractography recipes to identify the described tracts in other datasets. Original human data are available from the Human Connectome Project (www. humanconnectome.org). Raw and preprocessed human data from the Q2 and other public data releases are available from the Human Connectome Project (https://www.humanconnectome.org/ study/hcp-young-adult/data-releases/). Macaque data is made available through the PRIMatE Data Exchange (PRIME-DE; http://fcon_1000.projects.nitrc.org/indi/indiPRIME.html).

The following dataset was generated:

| Author(s) | Year | Dataset title | Dataset URL | Database and Identifier |
|---|---|---|---|---|
| Davide Folloni, Jerome Sallet, Alexandre A Khrapitchev, Nicola Sibson, Lennart Verhagen, Rogier B Mars | 2019 | Oxford WIN macaque PM | http://fcon_1000.projects.nitrc.org/indi/PRIME/oxford2.html | PRIMatE Data Exchange, oxford2 |

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
