## [Decision Letter]

Thank you for submitting your article "Two fiber pathways connecting amygdala and prefrontal cortex in humans and monkeys" for consideration by *eLife*. Your article has been reviewed by three peer reviewers, one of whom is a member of our Board of Reviewing Editors, and the evaluation has been overseen by Joshua Gold as the Senior Editor. The following individual involved in review of your submission has agreed to reveal their identity: Michel Thiebaut de Schotten (Reviewer #3).

The reviewers have discussed the reviews with one another and the Reviewing Editor has drafted this decision to help you prepare a revised submission.

Summary:

The reviewers agree that the paper, which details the course and composition of the uncinate fascicle and the amygdalofugal pathway, represents an important advance. It is well-written and clear, and should assist many future researchers in their studies of prefrontal-amygdala pathways in humans.

Essential revisions:

The reviewers raise a number of concerns that must be adequately addressed before the paper can be accepted. Some of the required revisions will require further analysis, but no additional experiments need to be undertaken.

1) One substantial issue with the paper as it currently stands is the conflation of bundles more broadly with amygdala-PFC connections specifically. It would be easy to look at the connectivity fingerprints shown in Figure 3 and assume that they reflect the strength of the connection *between the amygdala* and each PFC region, as opposed to what is represented generally within each bundle. For example, the uncinate fascicle, in addition to carrying amygdala-PFC fibers, also carries fibers between different portions of the ventral PFC. It also contains a large number of fibers moving between temporal lobe regions (STG, inferotemporal cortex) and PFC. These fibers have nothing to do with the amygdala. The amygdalofugal pathway as shown will include many passing fibers connecting with basal forebrain regions, among others. Although the Results section is relatively clear about this, other portions of the manuscript would make readers think that Figure 3 specifically shows amygdala-PFC projections, and the overall impression is confusing. The solution here is straightforward: do another set of analyses looking specifically at amygdala-PFC connectivities through each of these pathways. To the extent that the bundles are topographically organized, these should be somewhat different from the fingerprints shown in Figure 3. If they are not topographically organized (which perhaps the amygdalofugal pathways will not be), this may be a situation in which the limitations of the method will need to be made very clear.

2) All reviewers expressed concern that the text is overwritten in some places, and that prior literature is not properly addressed. Caveats and additional references to the literature are appropriate. The paper presents information that is not truly novel, but is importantly now ascertained using novel, in vivo techniques. The presence of the uncinate fasciculus and the ventral amygdala-fugal pathway and their complexity have been understood in some detail based on human post-mortem and monkey tract tracing studies for decades. In monkeys, several newer studies take detailed examinations of some of these fiber pathways in nonhuman primate. In humans, many conventional studies have documented the 2 pathways, albeit using different methods. Therefore, phrases such as 'little is known of structural wiring between the amygdala and PFC in humans…' is not really true. Thus, what's already known about the anatomy of the uncinate and the amygdalofugal pathways should be described in the Introduction. Reference to Nieuwenhuys et al. (2008) and Schmahmann and Pandya (2006) will not be enough for the lay reader.

3) The reader is left with the impression that these 2 tracts are the only ways that the amygdala communicates with the cortex. It is a bias of the method, and of course, the focus of the study. The idea that the amygdala has many additional cortical connections, sending fibers throughout the temporal lobe and occipital lobe via cingulum bundle and extreme capsule is worth mentioning. Furthermore, Lehman et al. (2011) show that some vmPFC fibers break off of the internal capsule to reach the stria terminalis, and travel to the amygdala that way. If the current paper wants to discuss all of the amgydala-PFC pathways, this is an important one that seems to be overlooked. (It should also probably be included in Figure 2.)

4) All reviewers also requested that the figures be substantially modified:

- There should be a few coronal slices added to Figure 2A-C so that we can see the entire rostral PFC-amygdala connection in both bundles. We should be able to see how they mix together rostrally, and how the amygdalofugal eventually moves to the amygdala caudally.

- The small images of the monkey and human brain do not have any visible information. I was interested in the differences in the VAF path in the human (to area 10 and 11) versus monkey, which seemed interesting. But it is difficult to appreciate the data. Later in the text (Discussion) the authors highlight the similarity of the VAF tracts into the medial frontal lobe. This is confusing. Figure 1 left panel: left and right hemispheres should be indicated or one of the 2 images flipped (I was confused at first).

- Figure 1 right panel. The image from the left is a duplicate from the one in the right panel. There surely is a way to combine tractography in macaque, tract tracing in macaque and tractography in humans a better way. Additionally, the anatomical background for human tractography is a T1w whereas the macaque is a DWI. This makes it harder to compare. Can you use a similar background?

- Figure 2. This figure is key and shows the anatomy of the amygdalofugal pathway in regards to the uncinate. The authors should delineate the amygdala in the two species. As it stands the illustration suggests the amygdalofugal is actually emerging from the caudate nucleus.

---

## [Author Response]

Essential revisions:The reviewers raise a number of concerns that must be adequately addressed before the paper can be accepted. Some of the required revisions will require further analysis, but no additional experiments need to be undertaken.

*1) One substantial issue with the paper as it currently stands is the conflation of bundles more broadly with amygdala-PFC connections specifically. It would be easy to look at the connectivity fingerprints shown in Figure 3 and assume that they reflect the strength of the connection* between the amygdala *and each PFC region, as opposed to what is represented generally within each bundle. For example, the uncinate fascicle, in addition to carrying amygdala-PFC fibers, also carries fibers between different portions of the ventral PFC. It also contains a large number of fibers moving between temporal lobe regions (STG, inferotemporal cortex) and PFC. These fibers have nothing to do with the amygdala. The amygdalofugal pathway as shown will include many passing fibers connecting with basal forebrain regions, among others. Although the Results section is relatively clear about this, other portions of the manuscript would make readers think that Figure 3 specifically shows amygdala-PFC projections, and the overall impression is confusing. The solution here is straightforward: do another set of analyses looking specifically at amygdala-PFC connectivities through each of these pathways. To the extent that the bundles are topographically organized, these should be somewhat different from the fingerprints shown in Figure 3. If they are not topographically organized (which perhaps the amygdalofugal pathways will not be), this may be a situation in which the limitations of the method will need to be made very clear.*

We thank the reviewers for this comment. We apologise for the confusion, the reviewers rightly point out that some of the terminology in the manuscript might have indeed conflated bundles with connectivity. In fact, this critical distinction between bundles and connectivity – and an appreciation of the limitations of diffusion tractography – have always been at the very heart of our study design choices, yet we had allowed a conflation to slip into our terminology. This was caused by our approach to use tractography in the best possible way to circumvent some of the just criticisms of the method that have occurred, but still find a way to link our results to previous tract-tracing results.

We have amended the manuscript substantially to deal with the issues raised by the reviewers. First, we have changed the Introduction and Discussion to clearly highlight the goal of the study and how we have tailored our approach in accordance with that goal. We now refer to our findings with a more consistent terminology throughout the manuscript to help differentiate between the concepts of bundles and connections. Second, to further clarify and highlight this distinction and our contribution to the literature, we have included additional analyses to investigate the extent to which bundles and areal connectivity show similar patterns. We will elaborate on both points below and describe the changes we have made in the manuscript.

The goal of our study was to investigate principles of anatomical connectivity between the amgydala and anterior temporal region (ATR) on the one hand and the prefrontal cortex on the other. In traditional tracer work, this has been done by means of injections into grey matter areas and subsequent observation which regions were reached by the tracer. Importantly, we build our work both on tracer studies that stain the neuronal cell bodies in grey matter (e.g. staining the source and termination areas of a connection), but also on tracer studies that additionally stain the axonal bundles through which the connections run. As pointed out by a series of critical studies, an area-to-area tracing approach is not feasible using tractography due to a number of issues, not in the least the presence of superficial white matter just underneath the grey matter border (Reveley et al., 2017) and the fact that brain area-to-brain area connectivity is too unconstrained to reliably pick up specific pathways. Therefore, we adopted an approach that put fiber bundles at the heart of the analysis. We and other groups have shown that by seeding in the core white matter of a fiber bundle it is possible to reliably track the bundle throughout its course (Thiebaut de Schotten et al., 2011; Mars et al., 2016; Mars et al., 2018). Placing the white matter bundle at the centre this way allowed us to investigate whether the bundles we identified followed the same course as those identified in earlier tracer studies and thus whether similar principles of organization were uncovered.

The downside of this approach is that it indeed does not immediately provide much information about areal connectivity. We therefore extended our analyses in two ways.

First, in the prefrontal cortex we test whether streamline counts of tractography seeded in either of the fiber bundles of interest reach the white matter around particular areas of interest. Although the seeding in bundles makes this procedure more reliable, it does not address gyral biases and other confounding effects in areas. However, by comparing patterns of connectivity across a range of regions – the so-called connectivity fingerprint – we are comparing like with like. Minor biases caused by gyrification will be the same for tractography seeded in the UF and in the AmF and will thus not bias our results.

Second, in order to investigate the difference between amygdala and temporal connectivity of the two bundles, we take the opposite approach. We seed in these areas and see whether we end up in the core of AmF and UF bundles. Again, we make the core of the bundle an essential step in the analysis, rather than focusing on area-to-area connectivity. This analysis showed that the principles of a lateral-medial organization also holds within the bundles, with temporal polar seeds leading to streamlines in lateral parts of the bundles and amygdala seeds leading to streamlines in medial parts of the bundles.

We hope these clarifications and additional analyses serve to both clear up the conflation of terms in the previous draft and substantially enhance the scope of the results.

The major changes to the manuscript reflecting these points are as follows:

Clarification of the goal and approach in the Introduction:

*“*To understand these computational roles requires an understanding of the anatomy underlying the brain systems involved (Marr, 1982). […] Therefore, we here apply this technique first in the macaque monkey, for which tracer results are available and comparison between the techniques can thus be established, and then to the human.”

New Materials and methods:

“We further asked how streamlines origination in either the amygdala or temporal pole were organized with respect to each other within the AmF and UF bundles. To this end, we performed a within-bundle connectivity-gradient analysis. […] A gradient ratio describing the relative occurrence of Amyg and TP streamlines was defined for the AmF bundle as:ratioAmF=AmFAmyg-AmFTPAmFAmyg+AmFTP, and for the UF bundle as:ratioUF=UFAmyg-UFTPUFAmyg+UFTP. In other words, these indices quantify the relative probability that streamlines coursing through the AmF and UF bundles have originated from either the amygdala (positive values) or the temporal pole (negative values).”

New Results:

“These results describe a preserved dichotomous organization of the AmF and UF along a predominantly medial-lateral axis across both species. […] A similar pattern is evident in the AmF, although it contained only very few streamlines originating in the temporal pole.”

Clarifications of approach and its limitations in the Discussion:

“It is important to point out that our use of diffusion-weighted MRI tractography necessitated some different protocol choices than a conventional tract tracing study would do. […] It also allows us to more easily create tractography recipes for the fiber bundles of interest that can then be shared with other researchers (e.g., Bramson et al., 2019).”

New Figure 4.

2) All reviewers expressed concern that the text is overwritten in some places, and that prior literature is not properly addressed. Caveats and additional references to the literature are appropriate. The paper presents information that is not truly novel, but is importantly now ascertained using novel, in vivo techniques. The presence of the uncinate fasciculus and the ventral amygdala-fugal pathway and their complexity have been understood in some detail based on human post-mortem and monkey tract tracing studies for decades. In monkeys, several newer studies take detailed examinations of some of these fiber pathways in nonhuman primate. In humans, many conventional studies have documented the 2 pathways, albeit using different methods. Therefore, phrases such as 'little is known of structural wiring between the amygdala and PFC in humans…' is not really true. Thus, what's already known about the anatomy of the uncinate and the amygdalofugal pathways should be described in the Introduction. Reference to Nieuwenhuys et al. (2008) and Schmahmann and Pandya (2006) will not be enough for the lay reader.

We thank the reviewers for these comments. We have now modified the text in the Introduction of the manuscript to tackle these issues. We now present a more complete overview of the literature as well as highlighting clearer what the extent and limits of previous studies was. It now reads as follows:

“To understand these computational roles requires an understanding of the anatomy underlying the brain systems involved (Marr, 1982). […] Therefore, we focus our investigation on the translatability of knowledge of macaque amygdala/anterior temporal-prefrontal fiber bundles to the human brain on these two tracts.”

3) The reader is left with the impression that these 2 tracts are the only ways that the amygdala communicates with the cortex. It is a bias of the method, and of course, the focus of the study. The idea that the amygdala has many additional cortical connections, sending fibers throughout the temporal lobe and occipital lobe via cingulum bundle and extreme capsule is worth mentioning. Furthermore, Lehman et al. (2011) show that some vmPFC fibers break off of the internal capsule to reach the stria terminalis, and travel to the amygdala that way. If the current paper wants to discuss all of the amgydala-PFC pathways, this is an important one that seems to be overlooked. (It should also probably be included in Figure 2.)

We want to thank the reviewer for giving us the chance to explain why we specifically focus on the Amygdalofugal Pathway (AmF) and the Uncinate Fascicle (UF) in the manuscript and aimed to disentangle their respective architecture. Indeed, we do not want to claim these are the only bundles that reach/leave the amygdala, rather we believe they are well suited to investigate the translation of organizational principles of amygdala/anterior temporal–prefrontal connectivity.

First, the reason why we did not include the Cingulum Bundle (CB) is because Heilbronner and Haber (2014) showed that the perigenual Cingulum Bundle carries amygdala fibers that first travel through the AmF before joining the CB in the white matter (WM) adjacent to the subgenual cortex and innervate anterior cingulate cortex and dorsal-medial cortical areas. The authors therefore show that the prefrontal segment of CB does not directly interconnect with the amygdala but it does so via the AmF. These results seem therefore consistent with what reported in our manuscript where we aim to describe and reconstruct tract that directly interconnect ATR and prefrontal regions. The authors also show that amygdala fibers can be also found in the temporal portion of the CB, leaving the amygdaloid nuclei caudally and joining this tract after briefly using the middle longitudinal fasciculus and inferior longitudinal fasciculus as a temporary conduit. However, Heilbronner and Haber show that such axons do not curve around the splenium of the corpus callosum, thereby not projecting to frontal lobe structures. Amygdala fibers, therefore, indirectly reach the CB only rostrally by running through the AmF first. Moreover, Lehman et al. (2011) suggested that lateral prefrontal granular areas use the external/extreme capsule complex briefly in the frontal lobe as a conduit to reach the AmF but not as a main bundle to connect with the amygdala. The extreme capsule is indeed a major association pathway primarily interconnecting a constellation of more dorsal-lateral temporal regions with rostral-lateral frontal areas (Pandya et al., 2015). For this reason, we did not include the external/extreme capsule complex within the bundles coursing between amygdala and prefrontal cortex. Also, the resolution of the diffusion imaging we used does not allow us to completely disentangle the extreme capsule from the external capsule and it is therefore likely that the former may include some fibers actually coursing in the latter.

Focusing specifically on amygdala-prefrontal cortex bundles, the amygdala has three main efferent bundles used by fibers to leave the structure and innervate rostral regions. As pointed out by the reviewer, these tracts are the AmF, the UF, and the Stria Terminalis (ST). However, previous tract tracing literature linked the fibers coursing in the ST primarily with amygdaloid, hypothalamic, septal, striatal structures, and the bed nucleus of stria terminalis, rather than with prefrontal projections (Nauta, 1962; Cowan et al., 1965; Russchen et al., 1985; Oler et al., 2017; De Olmos and Ingram, 1972). It is, however, true that Lehman et al. (2011) found that after injecting anterograde/bidirectional tracer in the caudal ventral-medial prefrontal cortex (PFC) some fibers were carried by the ST to the amygdala. Because of such contradictory results in the literature, we preferred focusing only on the tracts with established projection to the whole extent of the PFC, specifically to the rostral granular cortex (Wallis, 2011; Murray and Rudebeck, 2018).

Following the reviewer’s comment, we have however looked deeper into the amygdala-PFC connections of the ST. Lehman et al. (2011) specify that the ST carries some fibers between caudal PFC and amygdala. We have therefore run an additional control analysis and reconstructed the full extent of the ST in the human brain (see Figure 5—figure supplement 1 for seed area) as reported in Author response image 1 and also looked at its topographic organization relative to other surrounding tracts Figure 5A). Probabilistic tractography, however, cannot be constrained to only track monosynaptic connections and therefore these results cannot resolve the contradictory results about prefrontal ST fibers mentioned above. Our results show that some streamlines course within the ST in the striatal and subgenual WM (Author response image 1, top and middle rows). However, compared to the AmF and UF, ST fibers are sparsely present, if at all, in the WM rostral to the genu of the corpus callosum (Author response image 1, bottom row). Lehman and colleagues also show that ST axons course between caudal ventromedial PFC and amygdala by joining the WM adjacent the anterior limb of the Internal Capsule (IC) rather than through the WM here associated with the AmF. We show a similar course for the ST also in our control analysis reported in Author response image 1, showing that the ST axons occupies a WM portion medial to the anterior limb of the Internal Capsule but lateral to the Fornix (Figure 5A). Moreover, the ST shows also a specific topographic organization with respect to the AmF. Indeed, the two tracts show a dorsal-ventral organization with the ST always coursing in the WM adjacent to the bed nucleus of the stria terminalis and septum but dorsal to the AmF.

4) All reviewers also requested that the figures be substantially modified:- There should be a few coronal slices added to Figure 2A-C so that we can see the entire rostral PFC-amygdala connection in both bundles. We should be able to see how they mix together rostrally, and how the amygdalofugal eventually moves to the amygdala caudally.- The small images of the monkey and human brain do not have any visible information. I was interested in the differences in the VAF path in the human (to area 10 and 11) versus monkey, which seemed interesting. But it is difficult to appreciate the data. Later in the text (Discussion) the authors highlight the similarity of the VAF tracts into the medial frontal lobe. This is confusing. Figure 1 left panel: left and right hemispheres should be indicated or one of the 2 images flipped (I was confused at first).- Figure 1 right panel. The image from the left is a duplicate from the one in the right panel. There surely is a way to combine tractography in macaque, tract tracing in macaque and tractography in humans a better way. Additionally, the anatomical background for human tractography is a T1w whereas the macaque is a DWI. This makes it harder to compare. Can you use a similar background?

As requested by the reviewers, we have updated both Figure 1 and Figure 2.

In Figure 1, we previously used a T1-weighted image for both species and compared them with a modified version of Oler et al. (2017). The white/grey matter contrast in macaque monkeys and humans might appear inverted (or as if we used a T1-weighted image for the humans but a diffusion-weighted fractional anisotropy image for the macaques) because the macaque data were collected post-mortem. Post-mortem acquisition sequences are characterized by longer relaxation times of brain tissue, which causes the white/grey matter contrast to be inverted compared to standard in vivo acquisitions as used in the case of the human brain or in vivo macaque diffusion imaging data. We have now also rotated the images adjusted from Oler et al. (2017) along their vertical axis and registered our tractographic results to a species-specific templates with a similar white/grey matter contrast in both macaque and human brains. The same hemisphere is also now presented for both species. Moreover, following the reviewer’s suggestion, now each species is presented only once.

- Figure 2. This figure is key and shows the anatomy of the amygdalofugal pathway in regards to the uncinate. The authors should delineate the amygdala in the two species. As it stands the illustration suggests the amygdalofugal is actually emerging from the caudate nucleus.

In Figure 2 we now include 8 coronal slices (Figure 2A) instead of 3, representing the portion of cortex extending from the amygdala, through orbitofrontal cortex to the frontal polar areas. We show how the two tracts can be dissociated in the rostral temporal lobe/caudal prefrontal cortex and how they start intermingling together in the caudal dysgranular cortex and then mix together more rostrally in the granular orbital cortex rostral at the level of the genu of the corpus callosum. Moreover, we have highlighted the location of the Amygdala (A) in both species.